# Research Status of High-Purity Metals Prepared by Zone Refining

**DOI:** 10.3390/ma14082064

**Published:** 2021-04-20

**Authors:** Liang Yu, Xiaoan Kang, Luona Chen, Kun Luo, Yanli Jiang, Xiuling Cao

**Affiliations:** 1Key Laboratory of New Processing Technology for Nonferrous Metals & Materials, Ministry of Education, Guilin University of Technology, Guilin 541004, China; 2010054@glut.edu.cn (L.Y.); xakang93@163.com (X.K.); 18270899752@163.com (L.C.); 2Guangxi Scientific Experiment Center of Mining, Metallurgy and Environment, Guilin 541004, China; 3Collaborative Innovation Center for Exploration of Nonferrous Metal Deposits and Efficient Utilization of Resources, Guilin University of Technology, Guilin 541004, China; 4School of Materials Science and Engineering, Changzhou University, Changzhou 213164, China; luokun@cczu.edu.cn; 5School of Exploration Technology and Engineering, Hebei Geosciences University, Shijiazhuang 050031, China

**Keywords:** zone refining, high-purity metals, purification

## Abstract

The zone refining method is a physical method for effectively purifying metals. Increasing yield and reducing impurity content have always been the focus of its research. This article systematically summarizes the relevant research on the production of high-purity metals by zone refining, including mechanisms, parameter optimization, zone refining types, analysis methods, limitations, and future development directions, and it provides relevant theoretical foundations for the production of high-purity metals as well.

## 1. Introduction

High-purity metals (99.999% (5N) or higher) are widely used in modern electronic information, aerospace, defense, and military industries listed in Table 1. The development trend of modern science and technology requires high-purity or ultra-high-purity of metals, because some important characteristics of metals are affected by the type and amount of impurities in the matrix, and some characteristics are even masked by trace elements [1].

Metals are often purified by vacuum distillation technology [2], ion exchange technology [3], extraction technology [4], electrolytic refining technology [5], zone refining technology [6], and other single or combined technologies.

Among them, the zone refining technology has a wide range of applications; in addition, it is simple and easy to control, has no pollution, has a high product purity, and is suitable for the final stage of preparing high-purity metals. Since Pfann [7] first used the zone refining technology to purify germanium in 1955, it has been widely used in the purification of high-purity metals and the preparation of high-quality, multi-variety refractory single crystals. Zone refining is a technology of deeply purifying metals. Its essence is to use the difference in solubility of impurity elements in the solid and molten state of the main metal to precipitate the impurities or change the distribution of the impurity elements. It provides an effective and easy method for preparing high-purity metals. Theoretically, high-purity metals up to 8N can be obtained.

**Table 1 materials-14-02064-t001:** Application of high-purity metals.

Metal	Application
**In** [8]	ITO targets, CIGS solar cells, liquid crystal displays, etc.
**Sn** [9]	Packaging materials, integrated circuits, refractory materials, etc.
**Ni** [10]	Stainless steel, alloy steel, high-temperature structural materials, etc.
**Cu** [11]	Audio products, integrated circuits, fatigue-resistant cables, etc.
**Co** [12]	Magnetic materials, super alloys, electronic component targets, etc.
**Ti** [13]	Large-scale integrated circuits, decorative materials, etc.
**Ga** [14]	Semiconductor materials, solar cells, catalytic materials, etc.
**Ge** [15]	Integrated circuits, photovoltaic cells, infrared optical materials, etc.
**Te** [16]	Aerospace, atomic energy, electronics industry, etc.
**Al** [17]	Target materials, integrated circuit wiring, optoelectronic storage media, etc.

## 2. Zone Refining Mechanism

### 2.1. Basic Principles

Zone refining is a technology of deeply purifying metals [18]. The distribution of impurity elements in the solid and liquid phase of the bulk metal melt is determined by the thermodynamic properties of the system. Impurities exist as solid solutions in the main metal. Solid solution is generally due to the presence of metal B atoms in the crystal lattice of metal *A*, thereby forming a solid solution. Due to the presence of trace impurities, the melting point of the metal may decrease or increase. The extent to which the melting point decreases or increases depends on the content of impurities. Decreasing the melting point causes the solid solution to change from the molten state to the solid state, and the impurities migrate from the solid phase to the liquid phase. When the melting point increases, the opposite is true, and the impurities migrate from the liquid phase to the solid phase. For the binary system composed of *A* and *B* (the metal *A* contains impurities *B*), there is the following relationship [1,19,20]:(1)$${\Delta T}_{f}=\frac{{R(T}_{f}^{0}{)}^{2}}{\Delta {\bar{H}}_{fA}}x_{B1\;}[1-\frac{x_{B2}}{x_{B1}}].$$

In the formula, “1” represents the molten phase, “2” represents the solid solution phase; *x_A_*_1_ and *x_B_*_1_ are the concentrations of metal *A* and impurity *B* before the change, while *x_A_*_2_ and *x_B_*_2_ are the concentrations of *A* and *B* after the change; $\Delta {\bar{H}}_{fA}$ is the heat of dissolution when 1 mol *A* melts into the solution from a solid solution state, and its value is positive. Due to the low impurity content, it can be regarded as a constant; $T_{f}^{0}$ is the melting point of pure *A*; $T_{f}$ is the melting point of the solid solution at the mole fraction of *x_A_*_2_; $\Delta T_{f}{=T}_{f}^{0}-T_{f}$ is the melting point change value; *R* is the gas constant, and the value is 8.314.

Assume
(2)$$k_{0}=\frac{x_{B2}}{x_{B1}}.$$

If *k*_0_ < 1, that is *x_B_*_1_ > *x_B_*_2_; then, *x_A_*_1_ < *x_A_*_2_, and the concentration of *A* in the solid phase is greater than the concentration of *A* in the liquid phase. When $\Delta T_{f}\;$ > 0, the melting point rises, as shown in Figure 1b. If *k*_0_ > 1, that is, *x_B_*_1_ < *x_B_*_2_; then, *x_A_*_1_ > *x_A_*_2_, and the concentration of *A* in the solid phase is smaller than the concentration of *A* in the liquid phase. When $\Delta T_{f}\;$ < 0, the melting point drops, as shown in Figure 1a. In Figure 1, the upper part is the molten liquid zone; the middle is the solid–liquid two-phase equilibrium zone; the lower part is the solid phase zone.

In the case of *k*_0_ < 1 (see Figure 1b), the solid solution containing impurities at the phase point *P* is heated to the molten state *S* and then cooled to the temperature *T*; the first solid phase point to condense is *Q*, and the impurity content is reduced compared to the phase point *P*. Then, the solid phase at the phase point *Q* is heated and melted until it reaches the phase point *R*, after which it is condensed, and the impurity content in the obtained solid at the phase point *E* is reduced again, and the heating is repeated with condensation, the impurity content in the solid phase is continuously reduced to achieve the purpose of purifying metals, but the opposite is true for *k*_0_ > 1.

### 2.2. Analysis of Changes in Impurities during Zone Refining

The specific process of zone refining is shown in Figure 2a. The metal materials to be purified are placed in the tubular furnace; then, install a movable heating ring outside the tube (high frequency heating ring can be used).

Let the initial concentration of an impurity in the metal ingot be *C*_0_, which is locally melted into a melting zone; then, slowly move the heating ring to the right. The melting zone also slowly moves from the left end to the right end. The melted metal at the left end gradually solidifies again.

At this time, impurities with *k*_0_ < 1 are precipitated at the interface between the “resolidification zone” and the melting zone, and the concentration of impurities distributed in the liquid phase is greater than that in the solid phase.

Therefore, as the melting zone moves to the right, the impurities also move to the right, and finally, they are concentrated at the tail end, and the impurities of *k*_0_ > 1 mainly accumulate at the head end of the ingot. So, we repeat the above process many times; just like a broom sweeping the impurities to the two ends, the impurities are removed from both ends, purifying the middle of the ingot.

When using multi-melting zone refining, the advantages of zone refining can be seen. As shown in Figure 2b, a series of closely spaced heaters are used to melt into multiple melting zones in the ingot; after multiple zone refining, the impurity concentration distribution reaches a steady state or limits distribution.

Impurities do not diffuse in the solid phase, but the mass transfer rate in the liquid phase is very fast, and the concentration of impurities in the liquid phase is uniform. After a zone refining and purification, the distribution of impurities along the length of the ingot can be expressed by Formula (3) [19,21]:(3)$$C_{S}{=C}_{0}\;\left[1-\left(1-k_{0}\right)e^{-x\cdot k_{0}/l}\right].$$

*C_S_* is the impurity concentration at the distance *x* from the head end, and *l* is the length of the melting zone.

The two sides of Formula (3) respectively derive *x*:(4)$$C_{S}^{,}=\frac{C_{0}k_{0}\left(1-k_{0}\right)}{l}e^{\frac{-k_{o}x}{l}}\;>\;0.$$

It can be obtained that the *C_S_* moving direction along the melting zone is an increasing function, and it can be seen from Equation (3) that when *x/l*→∞, there is *C_S_/C*_0_→1, and the purification effect is not ideal at this time, so the value of *x/l* is not suitable if the width of the melting zone is fixed, and thus the ingot should not be made too long. The distribution of impurities along the ingot after the molten zone passes through the ingot is shown in Figure 3.

For the case of *k*_0_ > 1, the impurities in the melting zone are enriched in the solid phase. Since the impurities cannot move to the right with the melting zone in the solid phase, nor can they migrate to the left-side solidification zone. It is equivalent to distributing the impurities in the length of the rightmost *l* (melting zone width) of the ingot evenly within the range of the length of the left *x*-*l*, as shown in Figure 3b.

It can be seen from the combination of Figure 3 that for impurities containing *k*_0_ < 1 and *k*_0_ > 1 in the metal, in order to obtain a better purification effect, the middle part of the bar ingot should be a high-purity metal zone. The ratio of the length of the ingot to the width of the fixed melting zone (*L*/*l*) should not be too large, and the number of repetitions of melting has the best value, which is about *L*/*l*, and the cut length at both ends should be greater than the length of the melting zone *l*.

## 3. Influencing Factors and Optimization of Zone Refining

Since Pfann [22] published the pioneering work of zone refining, domestic and foreign scholars have conducted a series of research and discussion on the influencing factors of zone refining from a practical and theoretical perspective. These factors are mainly equilibrium distribution coefficient, zone refining rate, melting zone width, diffusion layer thickness, and zone refining times.

Yang [23] believes that the ingot length/area length ratio (*L/l*), the type and purity of the container, the composition of the ambient gas, and the degree of vacuum all need to be carefully studied; otherwise, it will become a limitation to achieve the required performance.

### 3.1. Balanced Distribution Coefficient

The equilibrium distribution coefficient *k*_0_ of impurities is the key factor that determines their migration, and it runs through the segregation purification. By observing the degree to which the equilibrium distribution coefficient *k*_0_ deviates from 1, it is generally possible to predict the purification ability of the zone refining process to the corresponding impurities. However, in the actual operation process, usually, the impurities have not had time to diffuse, and the temperature has changed greatly, causing the solid/liquid interface to advance greatly, forming a solid phase of new components; that is, non-equilibrium solidification actually occurs, and impurities are concentrated at the solidification interface [24].

Therefore, the equilibrium distribution coefficient *k*_0_ is not fixed, so the effective distribution coefficient (*k_eff_*) is introduced to improve the zone refining efficiency. The simplest method is to optimize *k_eff_*. The BPS equation is as follows [25,26]:(5)$$k_{eff}{=k}_{0}/\left[k_{0}+\left(1-k_{0}\right)e^{-f\Delta /D}\right].$$
where *f* is the zone melting rate, *D* is the impurity diffusion rate, and *δ* is the thickness of the boundary layer. As shown in Figure 4, when *f* ≫ *D*/*δ*, the rate of change of *k_eff_* increases rapidly until it approaches 1; when *f* ≫ *D*/*δ*, *k_eff_* tends to *k*_0_, and the melting efficiency at this time depends on the balance of specific impurities partition coefficient. In terms of purification effect, *k_eff_* is expected to be close to *k*_0_.

It can be clearly seen from Equation (5) that if the δ value is large, the *k_eff_* is also large, which is not conducive to zone refining.

Therefore, the *δ* value must be reduced. *δ* is directly affected by the mixing of the melt. During the zone melting process, measures such as rotation, external magnetic field, and oscillation stirring are often used. If the melt only has convection, *δ* is about 1 mm, which can be reduced to 0.1 mm by stirring. Stirring can be reduced to 0.01 mm [27].

### 3.2. Zone Refining Rate

The selection of the moving speed of the melting zone (that is, the zone refining rate) directly affects the purification efficiency, which is related to the production cost. Therefore, in the purification process, it is necessary to formulate an appropriate rate according to the characteristics of the material itself. When the melting zone movement rate *f* is very large, the solid phase crystallization rate increases accordingly, making it difficult to remove impurities in the unit cell, and the purification effect is not obvious; when *f* is small, the component diffusion can be fully carried out, and the impurities inside the crystal can be effectively removed, making the purification efficiency higher [28]. When multiple impurities are involved, the optimal zone melting speed is determined by the impurity with *k_eff_* closest to the host metal [29]. Prasad [28] optimized its process parameters when purifying tellurium and found that when the zone melting speed was 30 mm/h, it produced better results and obtained 7N pure tellurium material. Zhang [30] found that the melting rate has a significant effect on the zone melting process when purifying metallic tin. When the zone refining rate was reduced from 1.4 to 0.6 mm/min, the metal purity in the ingot increased from 99.99824% to 99.99906% after 10 times of refining. Wan [31] proposed an improved vacuum zone refining process. By modifying the effective distribution coefficient, the axial segregation of impurities was studied. When the zone refining rate was 1 mm/min, the aluminum content was greater than that of the 5N (99.9992%) sample.

When preparing single crystal materials, the use of a lower zone refining rate can effectively remove impurity elements inside the crystal and reduce crystal defects, but too low a zone refining rate will cause severe segregation of elements within the crystal and increase the concentration gradient of the component, which is not conducive to the preparation of single crystals; the use of a higher zone refining rate can improve the segregation of impurities and improve production efficiency, but it is not conducive to the elimination of gaseous impurities, which affects the stability of the solid/liquid interface, and it also affects the preparation of single crystals. Therefore, the choice of the optimal refining rate is to strike a balance between the formation of single crystals and the reduction of the degree of segregation [32].

### 3.3. Length of Melting Zone

The length of the melting zone is affected by many factors, such as thermal field, zone refining speed, crucible thermal conductivity, etc. In zone refining, the purity of the product obtained in the narrow melting zone is higher than that in the wider melting zone, but the impurity concentration of the narrow melting zone increases faster, making it difficult to remove impurities in the melting zone, which must be compensated by extending the purification time [33]. Therefore, in order to improve the purification efficiency, the actual production process generally adopts the operation method of using a wide melting zone and a higher zone refining rate at the beginning of purification and then changing to a narrow melting zone and a lower zone refining rate. Jun [34] purified industrial cerium by induction heating and found that the larger zone refining length in the early zone refining and the shorter zone refining length in the later stage can improve the segregation efficiency of solute during the zone refining process, especially for *k*_0_ far from the whole solute.

During the first zone refining, the impurity concentration of the melting zone will increase for *k*_0_ < 1; for *k*_0_ > 1, the impurity concentration will decrease. In order to meet the two situations at the same time, to obtain the best separation effect after the first zone refining, the melting zone must be as large as possible; that is, the length of the melting zone is the full ingot length. However, when the zone is melted a few times, see Figure 5. As the length of the melt zone increases, the “limit distribution” curve moves upwards, and the final purity that can be achieved decreases.

Ho [36] obtains the optimal melting zone length value for ten passes of zone refining. It can be seen from Figure 6a that the length of the melting zone increases with the increase of the distribution coefficient and decreases with the increase of the number of zone refining. Following this step for zone refining, a considerable separation effect can be obtained.

As shown in Figure 6b, when *k*_0_ < 1, the maximum solute removal rate decreases with the increase of the partition coefficient, and the opposite is true when *k*_0_ > 1. However, the maximum solute removal rate increases with the increase of zone refining times. Ghosh [37] calculated the zone melt length when meeting the maximum solute removal rate according to Ho’s numerical model. It can be obtained that as the number of smelting passes increases, the shorter zone melt length can be used to obtain the maximum solute removal rate [38]. Table 2 lists the zone refining length when the maximum solute removal rate is obtained per pass. Table 3 lists the purity grade of metal after zone refining.

### 3.4. Zone Refining Scans

The zone refining process needs to be repeated many times. As the number of times increases, the purification effect also increases, but after a certain pass zone melting, the impurity concentration distribution is close to the “limit distribution”, as shown in Figure 5.

At this time, in the process of driving the impurities to both sides, impurities on both sides will also diffuse to the middle. For the convenience of operation, the empirical formula for the number of zone refining times *n* can be expressed as *n* = *kL*/*l* (where *k* is the empirical constant, the value range is 1 to 1.5; *L* is the length of the ingot; *l* is the length of the melting zone); in actual operation, the frequently used condition is *L*/*l* = 10, taking *k* = 1.5; then, *n* = 15 [39].

### 3.5. Application of Current in Impurity Transmission

Applying a current field during the refining process (see Figure 7) can improve the segregation of impurities at the solidification interface through electromigration. Dost [40] conducted a numerical simulation on the process of transporting impurities in the cadmium zone refining system. The results show that the conductivity of cadmium is higher, which can enhance the migration rate of impurities in cadmium. The gas in Figure 7 is argon, which is used to prevent active metals from reacting with other gases.

### 3.6. Inclination

During the refining process in the horizontal area, the longitudinal section of the ingot will become tapered due to the mass transfer and even cause the melt to overflow. To avoid this phenomenon, the crucible should be inclined at an angle *θ* relative to the horizontal direction, as shown in Equation (6) [41]:(6)$${\theta =tan}^{-1}{\;\times \;2h}_{0}\;\left(1-\gamma /l\right)$$
where *h*_0_ is the original height of the ingot and *γ* is the ratio of solid and liquid metal density.

Magnetic field, gas flow rate, and melt stability will also have an important influence on the purification effect. In addition, the phenomenon of backflow in the melting zone will also affect the effect of zone refining. This is because the cooling rate of the upper and lower ingots differs greatly, and the ingot shrinks unevenly and bends upwards [42].

The experimental optimization of the above-mentioned zone refining parameters is not universal because they depend on the properties of the specific system, such as equipment specifications, ingot diameter, melt viscosity, and impurity distribution coefficient.

## 4. Types of Zone Refining

Based on the same principle, there are various zone refining technologies. There are two main types, refining in the suspension area and refining in the horizontal area, as shown in Figure 8.

The emergence of refining in the suspended area has greatly promoted the study of refractory metals. This method can not only effectively remove volatile metals and gas impurities in refractory metals, as well as avoid secondary pollution of metals in the refining process, but also effectively control the metal melt flow [43]. Table 4 lists the differences between the two methods.

## 5. High-Purity Metal Analysis Method

The purity testing of high-purity metals should be based on the actual application needs as the main standard, which is generally measured by the reduction method. From the development and application of metal purity analysis at home and abroad in recent years, it can be seen that in the determination of trace and ultra-trace metal elements, a basic mode can be attributed to the effective sample decomposition method, efficient separation and enrichment method, and simple, fast, and accurate instrument analysis method [1].

The analysis methods to determine the purity of high-purity metal materials are divided into chemical methods and physical methods, and they mainly consist of chemical analysis methods. Table 5 compares various chemical analysis methods.

It can be seen that GDMS can be directly analyzed by solid sample injection, with a wide measurement range, a variety of single detection elements, and high detection accuracy. It has obvious advantages in high-purity detection, but the equipment is expensive.

ICP-MS is more popular and has the advantages of speed, economy, and convenience.

NAA can be used for precious metal analysis and testing.

High-purity metal purity analysis is an important indicator to measure the production level of industrial science and technology. It plays an inestimable role in the development of high-purity metal production processes and the development of the materials science industry.

At the same time, it also follows the progress of science and economic development with increasing improvement, and the analysis of metal purity in the future will develop toward the following aspects [1]:(1)In terms of the number of measured elements, it will develop from single elements to multiple elements simultaneously or continuously.(2)In the analysis method, it will develop from offline/manual operation to online/automatic mode.(3)In terms of data collection and processing, the application of mathematical methods such as chemo-metrics, pattern recognition, expert systems, artificial intelligence, and neural networks will help to improve the integrity and accuracy of test data.

## 6. Numerical Simulation

Zone refining experimental research takes a lot of time. Therefore, many numerical models have been developed to effectively predict solute distribution or optimize experimental parameters as well as provide guidance for empirical experiments. Reiss [46] assumes that the solute partition coefficient *k*_0_ and the melting zone length *L* remain unchanged, establishes a differential equation between the concentration of each substance, and obtains a mathematical model of zone refining. Lee [47] proposed a model capable of simulating zone refinement and drawing conclusions. This model can pass the repeated pass method even when the zone travel speed and zone length are relatively large and thus cannot be properly purified. Cheung [48] combined a numerical model and genetic algorithm to establish an optimized melting zone length model to achieve maximum purification efficiency. Jun [34] uses induction heating to refine the industrial cerium in a cold crucible and introduces the numerical model of solute redistribution in the process of zone refinement into the genetic algorithm to search for the optimal zone length of different zone channels to improve the solute redistribution efficiency. The experiment and calculation results show that the genetic algorithm is a useful tool to obtain the optimal zone length during the zone refining process, but the stability of the molten zone should be maintained in the cold crucible.

Tan Yang [49] reported that the zone refining by combining finite element simulation data and experimental results analysis concluded that the zone refining experiment on the distribution of temperature field around the heat source with ellipsoid, semicircular canals is more suitable than square grooves for the purification of zone refining, as shown in Figure 9. The scope of the safety threshold value of indium under a given preheated temperature will greatly boost zone refining efficiency and does not destroy the balance of zone refining. In addition, Chen Luona [50] applied the oxidation refining-zone refining joint method to refining indium heat coupling finite element analyses, as shown in Figure 10. Here, argon gas flow in a horizontal transverse turn-back, along the long axis of the zone refining equipment in the opposite direction from exports, presents “up” in the zone refining equipment, forming annular flow. The model more accurately reflects the oxidation refining and zone refining temperature field as well as flow field distribution.

### 6.1. Iterative Modeling of Constant K

For different distribution coefficients and melting zone lengths, Bochegov [51] considered the limit distribution of impurities after purification by zone refining technology and proposed analytical solutions. It divides the ingot into three regions in the length direction, and the impurity distribution function is shown in Equation (7).
(7)$$C_{S\left(x\right)\;}=\{\begin{array}{c} {Ae}^{Bx},\;\;0\;\leq \;x\;\leq \;L-2l\\ \frac{{\mathrm{Ae}}^{B\left(L-2l\right)}}{S\left(L-2l\right){+l}^{k-1}e^{k}}S\left(x\right)+\frac{{Ae}^{B\left(L-2l\right)}}{S\left(L-2l\right){+l}^{k-1}e^{k}}l^{k-1}e^{\frac{k\left(L-l-k\right)}{l}},\;\;L-2l\;<\;x\;\leq \;L\;–\;l\\ \frac{{Ae}^{B\left(L-2l\right)}}{S\left(L-2l\right){+l}^{k-1}e^{k}}{\left(L-x\right)}^{k-1},\;\;L-l\;<\;x\;\leq \;L \end{array}$$

*A* and *B* are constants, and *B* is derived from known conditions:(8)$$B=\frac{k}{l}\left(e^{Bl}-1\right).$$

*A* is determined according to non-normalized conditions:(9)$$C_{s0}L=\int_{0}^{L-2l}C_{s1}\left(x\right)dx+\int_{L-2l}^{L-l}C_{s2}\left(x\right)dx+\int_{L-l}^{L}C_{s3}\left(x\right)dx.$$
*x* is the coordinate of the crystallization front relative to the starting point of the ingot, and *c_s(x)_* is the impurity concentration function on the coordinate *x* on the solid phase, where *C_s_*_0_ is the initial impurity concentration. *S* is determined according to the following expression:(10)$$S\left(x\right)=-\frac{k}{l}\sum_{n=0}^{\infty }{\left(\frac{k}{l}\right)}^{n}\frac{{\left(l_{0}-x-l\right)}^{k+n}}{k\left(k+1\right)\left(k+2\right)\cdots \left(k+n\right)}\;$$
where *n* is the number of zone refining.

The vertical axis corresponds to the ratio *C/C_s_*_0_ of the impurity concentration to its initial concentration, and the horizontal axis corresponds to the coordinate relative to the origin of the ingot. Thinner lines correspond to the distribution with a finite amount of passages with their numbers increasing as the lines approach the limiting distribution, as shown in Figure 11.

### 6.2. Iterative Modeling of Variable K

Spim [52] proposed a new iterative model that can be used to study the effect of zone melt length and distribution coefficient on refining efficiency, including the final impurity concentration distribution and the minimum zone pass required to reach the final state. In this model, by considering four different regions along the sample, the solute concentration distribution in each subzone of the melting zone is quantified. The formula is as follows:(11)$$C_{S(X)}^{n}=\{\begin{array}{c} k_{i}\left(\frac{dx}{Z}\right)\left(\overset{M-1}{\underset{q=0}{\sum }}C_{S\left(q\;dx\right)}^{n-1}\right),\;X=0\\ C_{S\left(X\;-dx\right)}^{n}+\cdots +\frac{k_{i}dx}{Z}\left[C_{S\left(X+Z-dx\right)}^{n-1}-C_{S\left(X-dx\right)}^{n}\right],\;0\;<\;X\;\leq \;1-Z\\ \frac{C_{0}}{Z}-\frac{dx}{Z}\left(\overset{M-1}{\underset{q=0}{\sum }}C_{S\left(q\;dx\right)}^{n}\right),\;1-Z\;<\;X\;<\;1\\ C_{0}-\frac{\sum_{q=0}^{M-1}C_{S\left(q\;dx\right)}^{n}}{M},\;X=1 \end{array}.$$

Cheung [53] established a database containing the changes in solids concentration (*C_S_*) and liquid concentration (*C_L_*) to obtain the variable *k*, so during the simulation process, for each *C_L_* in any melting zone, *k* can be easily accessed from the database. Comparing the two model simulations with constant *k* and variable *k* for aluminum refining, it can be concluded that using variable *k* in the simulation is closer to the experimental impurity distribution curve than considering constant *k*. The results show that the variable solute distribution method is used to simulate the impurity distribution in the refining process in different regions, which is closer to the experimental result *k* than the usual method.

Cheung [48] interacted with the numerical model through the AI (artificial intelligence, AI) search method to quantitatively determine the influence of the variable solute distribution coefficient *k* on the distribution of impurities in multi-pass purification by zone refining. The results show that the interaction between the numerical model and the AI search technology can provide the best set of melting zone lengths to achieve maximum purification efficiency. As shown in Figure 12, each node represents the length of a fusion zone and is linked to subsequent nodes of possible solutions, resulting in an explosive combination of expansion.

Zhang [54] carried out a zone melting process to purify Sb under the conditions of inert gas protection and heater movement rates of 2, 1, and 0.5 mm/min, respectively. By fitting the impurity concentration curve obtained from the experiment with the Spim model, the *k_eff_* values of Pb, As, and Fe impurities at each moving rate are retrieved. This helps to more accurately predict the time required for zone refining and achieve the efficiency of Sb zone refining.

### 6.3. Iterative Modeling Considering Diffusion Area

Nakamura et al. [55] proposed a numerical simulation model that divides the melting into two independent regions, namely the diffusion region and the stirring region, as shown in Figure 13a, and the vertical axis C represents the solid and liquid. There is an interface between them, and there are diffusion regions and stirring regions in the molten region. This numerical model introduces a transfer ratio q to illustrate the amount of solute transferred from the diffusion zone to the stirring zone during the diffusion process.

Figure 13b is a schematic diagram of the model calculation; the length of the melting zone is equal to am (mm). When the first segment is the diffusion area, the concentration is as shown in Equation (12):(12)$$C_{L\left(i\right)}^{\delta }{=C}_{L\left(i-1\right)}+\left(1-k\right)C_{L\left(i-1\right)}^{\delta }\;\times \;\left(1-q\right).$$

The new mixing zone consists of three parts: the distribution of the diffusion zone, the (*m−j*) segment with the concentration of *C_L_*_(*i*−1)_ in the previous agitating zone, and one segment of the new solid zone with the concentration of *C_(x)_*. Therefore, the average concentration in the stirring area *C_L_*_(*i*−1)_ becomes
(13)$$C_{L\left(i\right)}=\frac{\left(1-k\right)C_{L\left(i-1\right)\;}^{\delta }{\times \;q+C}_{L\left(i-1\right)}\;\times \;\left(m-j\right)+C\left(x\right)}{\left(m-j\right)},\;2\;\leq \;j\;\leq \;m-10.$$

Burton [26] established two important equations related to the distribution of solutes when there is a diffusion layer between the solid zone and the molten zone.
(14)$$\frac{C_{L}-C_{S}}{{\;C}_{L}^{\delta }-C_{S}}=exp\left(-f\delta /D\right)$$

Assume that the exponential factor in Equation (14) is the transmission ratio *q*.
(15)exp(−fδ/D)=q

The experimental values of *f* and *D* have been determined, and if the transfer ratio *q* is clear, the width of the diffusion region *δ* can be estimated. The advantage of this model is that when the *δ* of one impurity is determined by parameter fitting, the same *δ* value can be used to derive the distribution curve of other impurities.

### 6.4. Evaluation Modeling and Experiment

Since there are various experimental parameter combinations in the process of improving zone refining efficiency, it is very time-consuming and even unfeasible to study only through experiments.

In addition, refining systems in different regions have the best experimental parameters. Therefore, the application of the numerical model as described above is a good assistant to guide the experimental performance. In this way, the influence of each parameter on the refining result can be predicted, and the theoretically optimal parameter combination can be obtained in a short time and at low cost.

Therefore, experimental research is necessary to prove the accuracy of simulation prediction or to correct the model itself. It is recommended to combine simulation and experiment to study the refinement of metal regions.

In order to carry out zone refining and prepare high-purity metal indium, corresponding process improvement and equipment research and development are required. For this reason, our team Li Qing [56], Tan Yang [49], and Li Yicheng [57] designed the vacuum zone refining system for a multi-refining zone, whose structure is shown in Figure 14.

The vacuum zone refining/electromigration refining device is composed of stepping motor numerical control equipment, a high-purity argon gas purifier, a high-frequency power supply, an induction heating coil, an electromagnetic shielding device, a quartz furnace tube, and a vacuum pump. The quartz furnace tube is equipped with a cooling jacket, and a quartz boat is placed inside to carry indium. In the process of zone refining, the width of the melting zone is controlled by the output of high-frequency power, while the moving speed of the melting zone is precisely controlled by the numerical control table of the stepping motor. The general speed is 3–30 mm/hour. An electromagnetic shielding device is used to prevent electromagnetic interference between the melting zones. The atmosphere during the refining process is controlled jointly by the diaphragm vacuum pump and the high-purity argon gas purifier.

First, the diaphragm vacuum pump pumps the vacuum up to about 0.06 MPa for 20 min. Then, the air valve of the vacuum pump path is closed, and the air valve leading to ultra-high-purity argon gas is opened for argon gas (the pressure of argon gas is always slightly higher than 1 atmospheric pressure). The above operation was repeated 3 times to complete the air exchange process, and the high-purity indium was prepared in the area melting/electromigration process under the protection of ultra-high-purity argon gas. The vacuum zone refining/electromigration joint method for high-purity indium extraction can be used without high vacuum under the condition of the preparation of high-purity 6N indium, and it can significantly reduce the technical threshold and gas raw material costs, indicating that the technology has obvious industrial application prospects.

In order to solve the problem of small processing capacity of metal indium in the zone refining process, Li Yicheng et al. [57] adopted the method of simultaneous heating of multiple furnace tubes, which doubled the processing capacity of the zone refining process. Figure 15 shows the working picture of the vacuum refining/electromigration high-purity indium refining plant in the single refining zone of 4 furnace tubes. At present, the small unit of the laboratory has been able to complete the continuous treatment of 100 kg indium per year. The whole process has a high energy utilization rate, with no waste water, waste gas, waste residue discharge, and environmental friendliness, and 6N indium can be obtained.

Li Qing [56] and Chen Luona [50] designed a new vacuum zone refining/electromigration high-purity indium refining device designed by Tan Yang [49] with multiple furnace tubes. The heat of an alumina ceramic heating sheet was introduced into the graphite sheet as a constant temperature heat source, as shown in Figure 16.

## 7. Existing Problems and Prospects

After nearly 70 years of development, the zone refining technology has made great progress in both process research and application field expansion. The process is becoming more mature, but there are still problems in the following aspects [1,27]:(1)The requirements for raw materials are high. The purity of the raw material rod must meet certain requirements, and the gap impurities C, H, O, etc. contained in it must be controlled within a certain range, so as to avoid the power and temperature fluctuations caused by gas impurities in the zone refining process, which affects the quality of the material.(2)For impurities with a balanced distribution coefficient close to 1, the content of the raw material rods must be strictly controlled so as not to affect the final performance of the materials: for example, magnesium, calcium, iron, and antimony in bismuth; lead, magnesium, silicon, and aluminum in indium.(3)The influence of side effects. Side effects include the evaporation of high vapor pressure impurities in the melting zone caused by agitation, temperature increase, or low pressure inert gas flow through the melting zone, due to chemical reactions between impurities (such as between carbon and sulfur, hydrogen, or oxygen), and the slagging process often brings complex problems that are difficult to predict. Sometimes, in order to remove an impurity, it is necessary to add a second reaction element to complete, and the latter is selected to generate a compound with the former. During zone refining, it is easier than removing certain impurities alone.(4)The bar size is limited. The emergence of refining in the suspension zone solves the crucible pollution existing in the conventional zone refining, making the zone refining technology develop rapidly. However, due to gravity, the technique requires that the size of the raw material rod must be controlled within a certain range to obtain a stable refining zone.(5)Low production efficiency and high cost. The low speed and multiple refining required by the zone refining greatly extend the production time and reduce the production efficiency; after each zone refining, the first and last ends of the bar need to be removed, which reduces the utilization rate of raw materials and increases the production cost.(6)The development speed of product testing and analysis technology lags behind. With the improvement of purification technology, a variety of highly sensitive impurity analysis methods came into being, but these methods have certain limitations, which restrict the development of zone refining technology.(7)The large-scale application of advanced zone refining technology is limited. The research on zone refining technology has made great progress, and it has been used in industrial production in certain fields, but some of the deficiencies have hindered its large-scale application: the suspension zone refining technology equipment investment is expensive, and raw material storage container selection is more difficult. Moreover, it is susceptible to secondary pollution during operation, and the optimal production process parameters are easily changed with the material.

In recent years, the market demand for high-purity metals has continued to expand, and zone refining has become the preferred method for preparing such materials, with broad application prospects. In view of the current market demand, the focus of research on zone refining at this stage should be shifted to the following aspects [27,54].

(1)Combining zone refining technology with other purification technologies, developing an ideal purification method combining multiple technologies, such as electromigration/zone refining, vacuum degassing-zone refining, to effectively remove gas impurities and ensure the stability of subsequent zone refining proceed to obtain higher purity materials.(2)Upgrade the zone refining equipment, increase the degree of automation and improve the zone refining technology to obtain a stable and simple operation process, improve production efficiency, and reduce production costs.(3)Improve trace element analysis technology to make simultaneous or continuous determination of trace and ultra-trace multi-elements. In addition, the industrialization of zone refining technology is also the main development direction in the future. With the deepening of research, zone refining technology will definitely develop toward the industrialization of low cost, practicality, high efficiency, high reliability, and high complexity.

## Figures and Tables

**Figure 1 materials-14-02064-f001:**
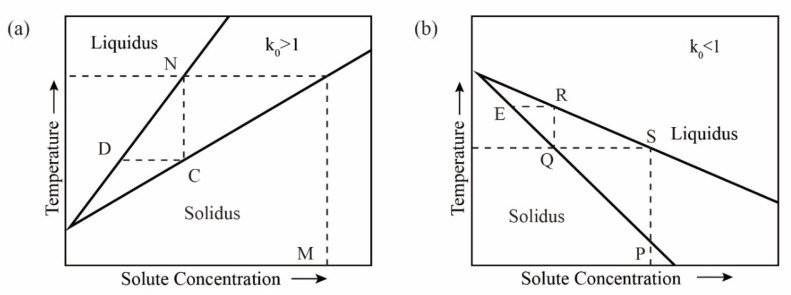
Binary system phase diagram; (**a**) *k*_0_ > 1; (**b**) *k*_0_ < 1.

**Figure 2 materials-14-02064-f002:**
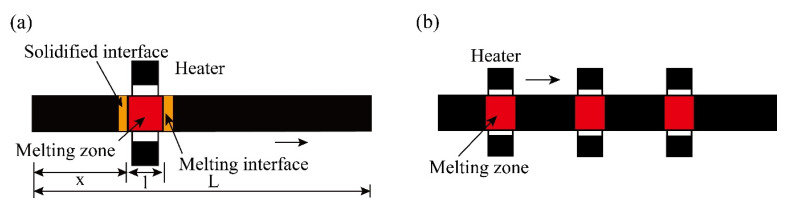
Specific process of zone refining; (**a**) Single-pass zone refining; (**b**) Multi-pass melting zone refining.

**Figure 3 materials-14-02064-f003:**
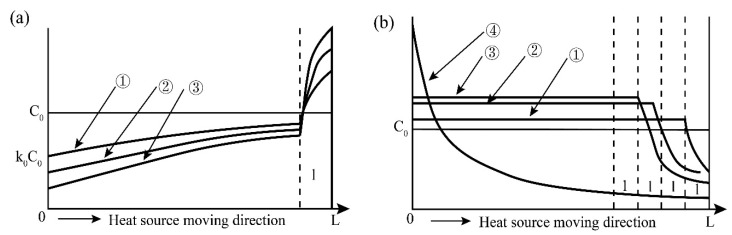
Distribution of impurities along the ingot after the molten zone passes through the ingot; (**a**) Distribution curve of multiple zone molten impurities along the ingot when *k*_0_ < 1; (**b**) Distribution curve of multiple zone molten impurities when *k*_0_ > 1.

**Figure 4 materials-14-02064-f004:**
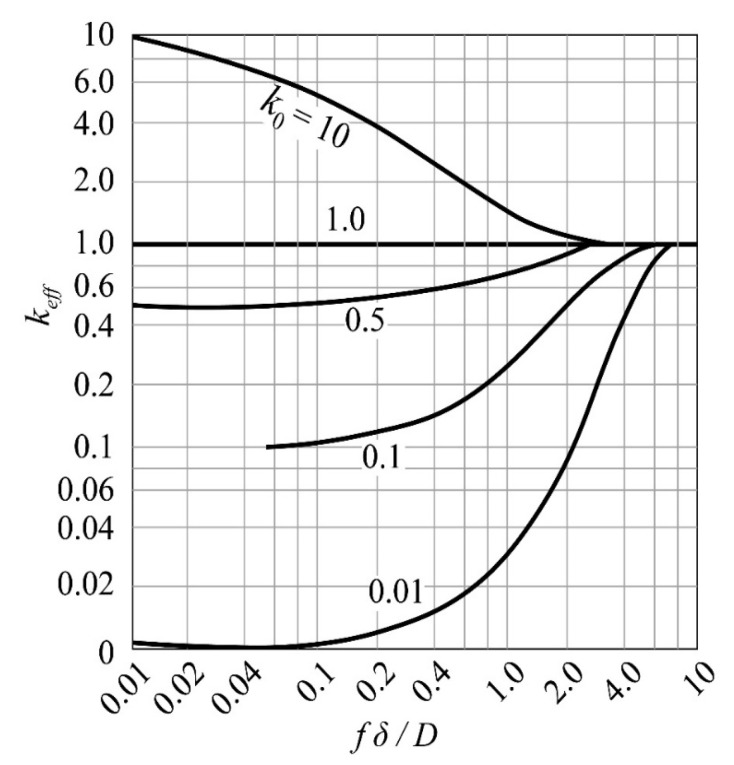
Relationship between *k_eff_* and *fδ/D.*

**Figure 5 materials-14-02064-f005:**
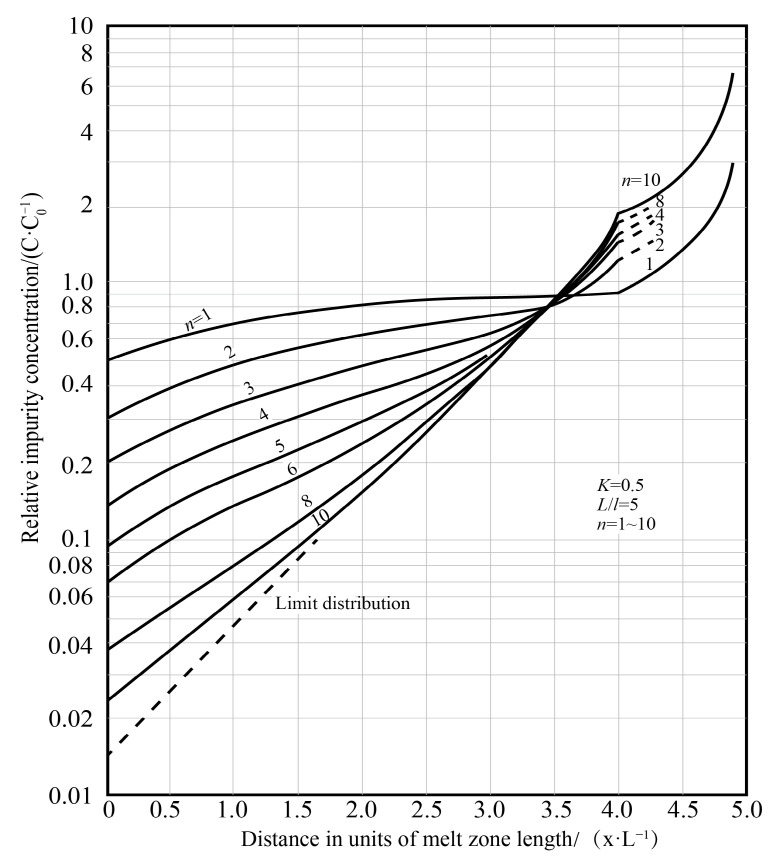
Relationship between impurity concentration and number of zone refining passes [35].

**Figure 6 materials-14-02064-f006:**
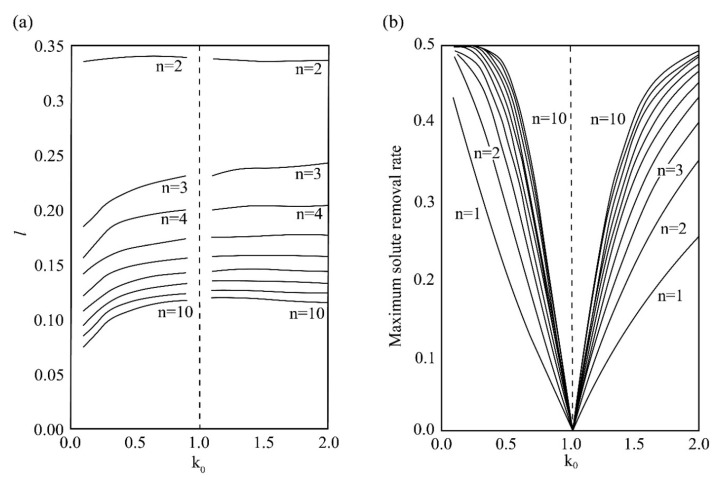
(**a**) Optimal zone refining length of 1–10 passes; (**b**) Maximum solute removal rate during multi-pass zone refining.

**Figure 7 materials-14-02064-f007:**
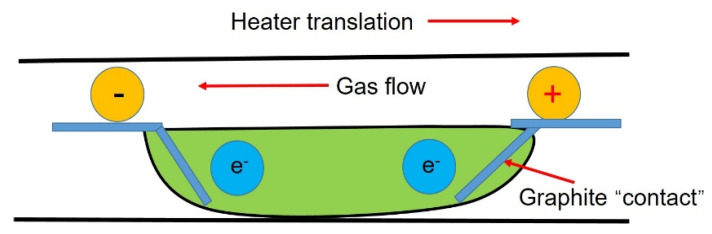
Schematic diagram of the zone refining system under applied current.

**Figure 8 materials-14-02064-f008:**
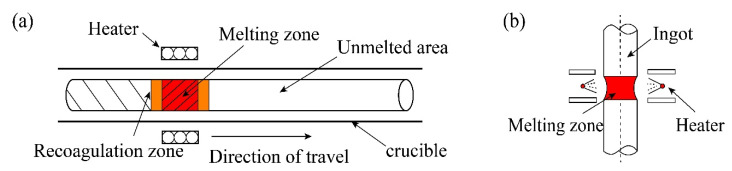
(**a**) Horizontal zone refining technology; (**b**) Suspended zone refining technology.

**Figure 9 materials-14-02064-f009:**
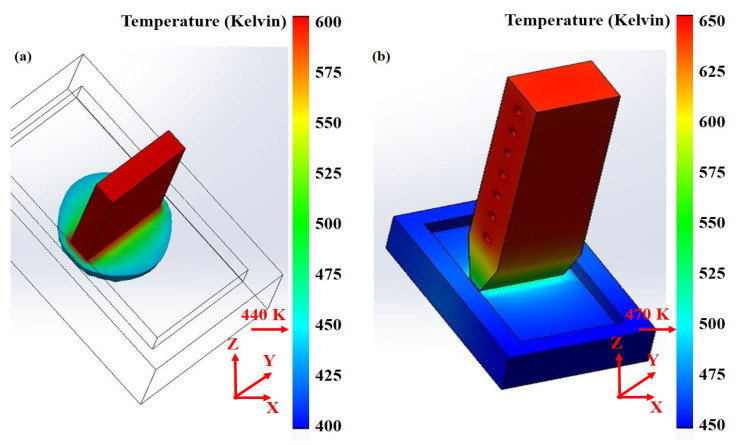
Temperature cross-section: (**a**) SiC thermal conductor, (**b**) graphite electrothermal assembly.

**Figure 10 materials-14-02064-f010:**
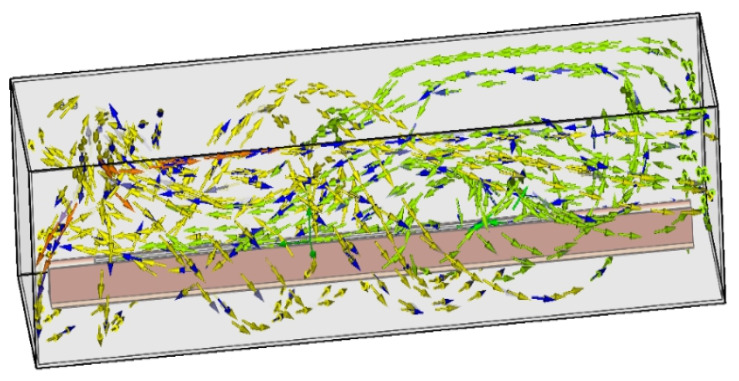
Flow diagram of argon gas flow in the zone refining equipment.

**Figure 11 materials-14-02064-f011:**
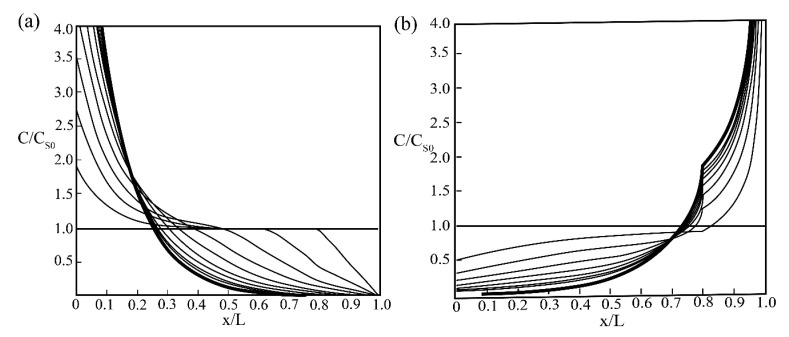
Limiting distribution for the case of molten zone size *l* = 0.2 and impurity distribution factor (**a**) *k* = 2 and (**b**) *k* = 0.5.

**Figure 12 materials-14-02064-f012:**
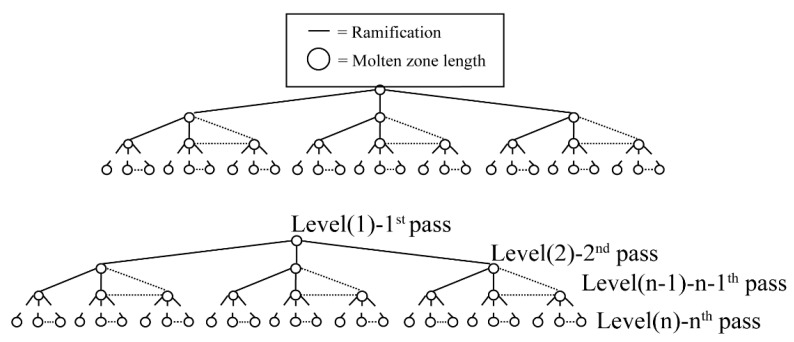
Representative tree of possibilities.

**Figure 13 materials-14-02064-f013:**
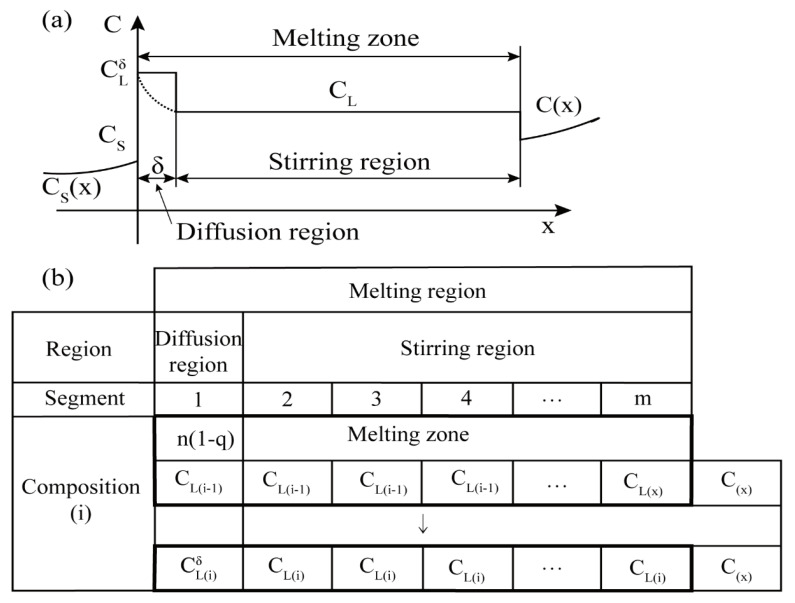
(**a**) Schematic diagram of the melting zone; (**b**) Schematic diagram of the computation model.

**Figure 14 materials-14-02064-f014:**
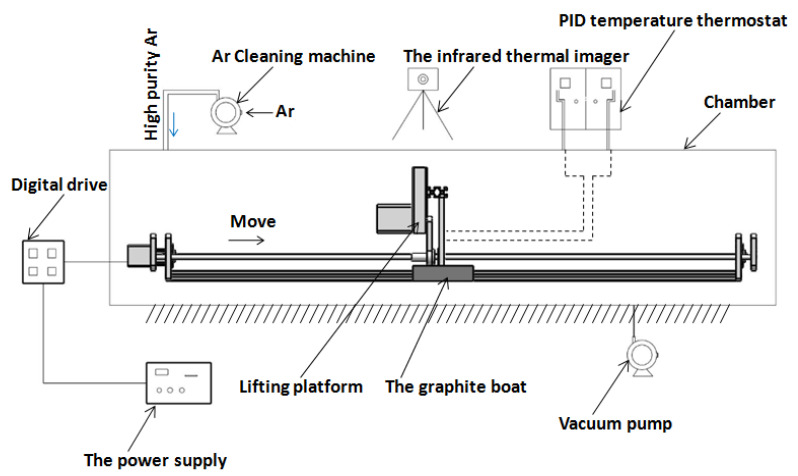
Schematic diagram of vacuum refining-electromigration high-purity indium refining plant in multi-refining zone.

**Figure 15 materials-14-02064-f015:**
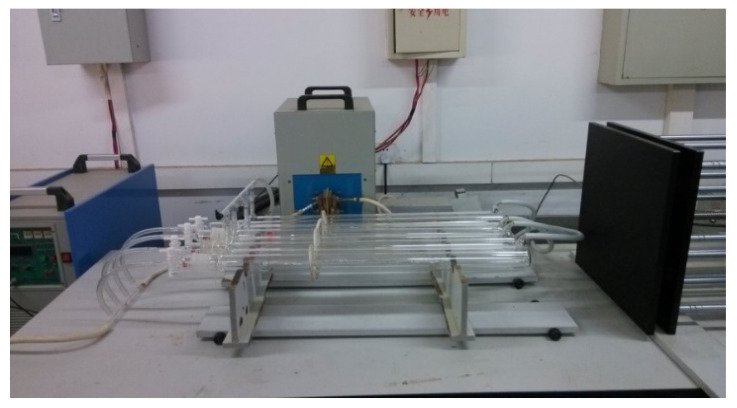
Vacuum zone refining/electromigration high-purity indium refining device designed with multiple furnace tubes.

**Figure 16 materials-14-02064-f016:**
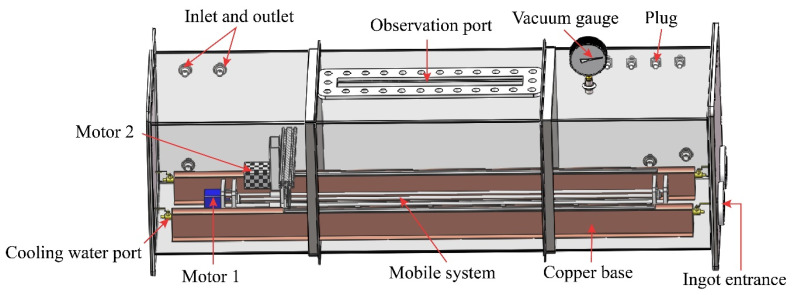
Zone refining equipment diagram.

**Table 2 materials-14-02064-t002:** Zone refining length when the maximum solute removal rate is obtained per pass.

Zone Refining Times	1	2	3	4	5–8	9–19		≥20
Normalized melting zone length	1	0.35	0.25	0.2	0.15	0.1		0.05

**Table 3 materials-14-02064-t003:** The purity grade of metal after zone refining.

Elements	Al	As	Cd	Cu	Fe	Mg	Ni	Pb	Ga	Si	Sn	Te	Zn	Ag
Purity grade	9N	9N	7N	10N	9N	8N	7N	7N	7N	9N	7N	7N	7N	8N
**Elements**	**G** **e**	**In**	**Ce**											
Purity grade	10N	7N	7N											

**Table 4 materials-14-02064-t004:** Differences in zone refining technology [43].

Category	Heating Method	Advantage	Disadvantage	Scale
Floating zone refining	Electron beam heating, induction heating, plasma heating, light heating	The ingot does not touch the container, so the product purity is high, and the equipment occupies a small space.	The melting zone is supported by surface tension, so controlling the shape and stability of the melting zone is the key, and the output is low.	Small batch
Horizontal zone refining	Induction heating, resistance heating	Simple equipment, continuous purification of multiple melting zones, easy loading and unloading of materials, easy identification of interfaces, and the total length of ingots can be increased or decreased as needed	Large footprint	Batch

**Table 5 materials-14-02064-t005:** Comparison of various analysis methods for measuring metallic elements.

Analytical Method	Sampling Method	Measuring Range (g/g)	Explanation
GDMS [44]	Solid	Constant~10^−12^	The matrix effect is small, the sample processing is simple, it can avoid pretreatment pollution and loss, it is fast and efficient, the detection limit is low, and standard samples can be used, but the price is expensive. The discharge is stable, and the analysis sample can be peeled layer by layer for surface and depth analysis. The self-absorption effect is small. Multiple elements can be determined simultaneously.
ICP-MS	Liquid	10^−6^~10^−12^	High resolution and sensitivity, a wide measurement range, a need to try to eliminate the matrix effect and isotope interference, and the sample processing process is longer.
GF-AAS	Solid	10^−6^~10^−9^	The number of single analysis is small, the analysis range is narrow, and it is not suitable for high melting point metals.
ICP-AES [45]	Liquid	Constant~10^−6^	Fast speed, high sensitivity, small matrix effect, low detection limit, and serious spectrum interference and matrix interference. Multi-element simultaneous analysis method with the widest range of analysis elements and the largest content span.
TXRF	Solid	Constant~10^−12^	Fast, simple, and economical non-destructive testing, which can perform multi-element analysis at the same time, but it can only perform surface analysis, and the detection limit is low.
NAA	Solid, liquid	10^−6^~10^−13^	High sensitivity, high accuracy and precision, and sensitivity varies from element to element. It can only measure the content of elements. It requires a small nuclear reactor and has the risk of nuclear radiation. Less sampling, non-destructive analysis. The detection equipment is expensive.
NTIMS	Liquid	10^−10^~10^−14^	It has a lower detection limit than ICP-MS, can be used for isotope age studies, and can accurately measure Re and Os.

## Data Availability

Data sharing is not applicable for this article.

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
