# Peer review of "Research Status of High-Purity Metals Prepared by Zone Refining"

_materials, 2021, doi:10.3390/ma14082064_

Round 1
Reviewer 1 Report
Research Status of High-purity Metals Prepared by Zone Refining is very interesting and important review paper. Some additional minor improvements are required:
Page 5: When the zone refining rate was reduced from 1.4 mm/min to 0.6 mm/min, the metal purity in the ingot increased. What is minimal length of the Zone refining (melting Zone)?
Page 7: Ghosh [37] calculated the zone melt length. Is the Zone melt legth depending on the smelting Point and type of smelted metal?
Page 8: At Figure 7 we can see "gas flow"? What is origin and type of gas? What is role of gas flow?
General Questions:
- Is this Technology suitable for refining of rare earth Elements and all nonferous metals
- What is maximal quantity of refined metal per one hour?
- What is small scale Zone refining (see TAble 4 at Page 8)
Author Response
Dear reviewers:
Thank you for your letter and the reviewers’ comments on our manuscript ID materials-1050672 entitled " Research Status of High-purity Metals Prepared by Zone Refining".
Those comments are very helpful for revising and improving our paper, as well as the important guiding significance to other research.
We have studied the comments carefully and made corrections which we hope meet with approval. The main corrections are in the manuscript and the responds to the reviewers’ comments are as follows (the replies are highlighted in red).
Most sincerely,
Liang Yu
4.08 , 2021
Review1
Page 5: When the zone refining rate was reduced from 1.4 mm/min to 0.6 mm/min, the metal purity in the ingot increased. What is minimal length of the Zone refining (melting Zone)?
Answer: Thanks for your very thoughtful suggestion. The length of melting zone is determined by purification equipment and type of metals.
Page 7: Ghosh [37] calculated the zone melt length. Is the Zone melt legth depending on the smelting Point and type of smelted metal?
Answer: Yes, the length of the melting zone depends on many factors, including the smelting point,type of the smelted metals.
Page 8: At Figure 7 we can see "gas flow"? What is origin and type of gas? What is role of gas flow?
Answer: The gas in Figure 7 is argon, which is used to prevent active metals from reacting with other gases.
General Questions:
- Is this Technology suitable for refining of rare earth Elements and all nonferous metals
Answer: Zone refining technology is suitable for semiconductors, rare earth metals and refractory metals.
- What is maximal quantity of refined metal per one hour?
Answer: The amount of refined metal depends on the zone refining equipment. Industrial equipment can smelt about 10 ~ 50 kg metal per hour
- What is small scale Zone refining (see Table 4 at Page 8)
Answer: Small scale zone refining means that Small scale experiments were carried out in the laboratory, but industrial production cannot be carried out.

Reviewer 2 Report
1-Language need to polish before consideration for publication.
2- It can be worthwhile if authors can add a section of Knowledge gap for preparing high purity metals by zone refining.
3-Authors should add a table of different categorize either s, p, d and f block elements or alkali metal, alkaline earth metals, transition metals and mention the purity scale.
Reviewer 3 Report
The review article "The state of research on high-purity metals obtained by zone refining" presents an important problem of metal refining methods. Due to the fact that nowadays there is a demand for metals with special properties the operations of their purification become very important. For this reason, I find the topic of the article interesting. . In characterizing the zonal refining process, the authors considered such factors as the equilibrium decomposition coefficient, the zonal refining rate, the width of the melting zone, the thickness of the diffusion layer, and the zonal refining times. In addition, the paper discusses zone refining methods and the methods used to determine the content of impurities in the refined metals.
The bibliography contains 57 items which is not a large number for a review article. However, it contains all the main items in the subject matter of the article and can be considered sufficient.
The article layout is legible. There is not classical summary in the article. Given the review nature of the study, it can be considered that such a role is played by the chapter 7 ”Existing problems and prospects”.
1) Table 3 - Authors should unify the description of the columns, i.e. the usage of capital letters 2) Table 4 - the description of 3rd column is divided 3) Figure 9 - the resolution of the figure is poor 3) Figure 13b and Figure 14 - the size of the font is too low
Reviewer 4 Report
This article systematically summarizes the relevant research on the production of high-purity metals by zonal refining, including mechanisms, parameter optimization, types of zone refining, analysis methods, limitations and future developments, and provides relevant theoretical foundations for high-purity metal production.
The article takes the form of a comparative study with our own research in the field of zonal metal refining. It has a very good level. The use of numerical modeling is beneficial.
The aim is to apply the method of zonal melting in industrial production. This will be one of the trends of development in the future.
Author Response
Thanks for your very thoughtful suggestion. We have revised it.
